# Production of Graphene Stably Dispersible in Ethanol by Microwave Reaction

Alberto Martis [1,2,*], Marco Fontana [1,2], Mara Serrapede [1,2], Stefano Bianco [2], Angelica Chiodoni [1], Candido Fabrizio Pirri [1,2] and Sergio Bocchini [1,2,*]

1 Center for Sustainable Future Technologies (CSFT)@Polito, Istituto Italiano di Tecnologia, Via Livorno 60, 10144 Torino, Italy
2 Department of Applied Science and Technology—DISAT, Politecnico di Torino, Corso Duca degli Abruzzi 24, 10129 Torino, Italy
* Correspondence: alberto.martis@iit.it (A.M.); sergio.bocchini@polito.it (S.B.); Tel.: +39-388-1712571 (S.B.)

**Abstract:** Graphene is a 2D carbon material with peculiar features such as high electrical conductivity, high thermal conductivity, mechanical stability, and a high ratio between surface and thickness. Applications are continuously growing, and the possibility of dispersing graphene in low-boiling green solvents could reduce its global environmental impact. Pristine graphene can be dispersed in high concentration only in polar aprotic solvents that usually have high boiling points and high toxicity. For this reason, the oxidized form of graphene is always used, as it is easier to disperse and to subsequently reduce to reduced graphene oxide. However, compared to pristine graphene, reduced graphene oxide has more defects and has inferior properties respect to graphene. In this work, the polymerization of (diethyl maleate derivate) on graphene obtained by sonication was performed in a microwave reactor. The obtained material has good stability in ethanol even after a long period of time, therefore, it can be used to deposit graphene by mass production of inks or by casting and easy removal of the solvent. The thermal annealing by heating at 300–400 °C in inert atmosphere allows the removal of the polymer to obtain pristine graphene with a low number of defects.

**Keywords:** graphene; microwave reaction; dispersion





## 1. Introduction

Graphene, a two-dimensional carbon lattice, has received tremendous attention due to its excellent mechanical, thermal, and electrical properties [1–4]. Graphene has an exceptional carrier mobility of up to $2 \times 10^5$ cm$^2$ V$^{-1}$ cm$^{-2}$ [5,6] and has become a golden candidate for printed flexible electronics. Graphene is a material that has several fields of application and many different techniques have been used to produce and to prepare dispersion to be used at research and at commercial level.

Graphene can be dispersed at relatively high concentration in polar, aprotic solvents such as n-methyl pyrrolidone (NMP) and dimethyl formamide (DMF) [7]. However, these solvents are toxic, have a high boiling point and their removal is difficult and expensive. The production of stable graphene dispersions in low-boiling solvents would allow the usage with techniques such as inkjet printing or electrospinning, or to simplify many existing processes where solvent removal is necessary. [7]

A good solvent candidate is ethanol, which has a boiling point of 78.1 °C, but, at the same time, it is challenging to prepare stable graphene dispersion because of its low surface energy (22.1 mJ m$^{-2}$) [7–9]. Several approaches have been tried to achieve effective exfoliation in presence of acceptable concentrations; the most common method to disperse graphene in polar solvents is based on oxidation of graphite to prepare graphene oxide (GO). GO can be dispersed easily in polar solvents such as water and can be chemically reduced to reduced graphene oxide (rGO). However, rGO is intrinsically different from pristine graphene containing a high number of defects [10,11].

Gomez et al. have demonstrated that it is possible to exfoliate graphene in ethanol and that the reached amount depends exclusively on the graphite amount. The final concentration, however, does not exceed 20 µg mL$^{-1}$ [12]. Other methods concern the exchange of solvent from NMP to ethanol with more cycles of dispersion–filtration of graphene; in this case, the maximum graphene concentration is 0.4 mg mL$^{-1}$ [13].

The most widely used method for dispersing graphene in polar solvents is by using surfactants, since these increase surface energy. This feature is dependent on the amphiphilic nature of the component. There are two main families: ionic surfactant classified in organic salts, such as sodium dodecyl sulphate (SDS) or sodium deoxycholate (SDOC) [14–16], and inorganic salt, such as $(NH_4)_2SO_4$, $Na_2SO_4$, $K_2SO$, with the use of electrochemical exfoliation methods [17,18]. Non-ionic surfactants are mainly polymers or amphiphilic molecules such as Tween 80, TritonX-100, [19] Polyvinil pirrolydone (PVP) [19–21] and also amphiphilic proteins [22]. Liang and Hersam have also used the combination of polymer, exfoliation via sonication, and solvent exchange to enhance the concentration in polar solvent reaching a concentration of graphene of 1 mg mL$^{-1}$ [23].

Aromatic "π–π stacking" molecules with ionic or polar moieties as surfactant is another typical method for dispersing graphene. Pyrene derivates, such as 1-pyrensulfonic acid sodium salt, were studied by Parviz et al. [24], showing the ability to disperse graphene in water up to a concentration of 1 mg mL$^{-1}$ with a mean zeta potential of $\pm 30$ mV. Sodium cholate allows a good dispersion of graphene up to 0.3 mg/mL [8]. With hydrophobins, a class of amphipathic proteins in which the apolar portion can interact with the surface of graphene and the polar component can interact with the solvent, the concentration in water obtained is circa 2 mg mL$^{-1}$ [22].

Vinyl monomers such as styrene and methyl acrylate were successfully bound to the basal plane of graphene and polymerized to produce block copolymers with hydrophilic monomers such as ethylenoxide, allowing a good water dispersion [21,25]. The formation of a surface coating using nylon is another interesting possibility developed by Syradas et al., in which the nylon-coated graphene shows enhanced stability in water [24,26].

Polyvinyl pyrrolidone (PVP) is a non-toxic and not ionic polymer that has shown good properties as a surfactant to disperse graphene in a polar solvent with high concentration. The presence of the polymer on the surface of the carbon material with the polar moieties on the external part gives a dispersion of 0.8 mg mL$^{-1}$ in ethanol and about 0.42 mg mL$^{-1}$ in water [27].

An interesting work published by Lu Zhang et al. shows the dispersibility of graphene in low boiling point solvents such as acetone and methanol using edge-carboxylated graphene quantum dots (ECGD) as a surfactant. This material interacts with the graphene π–π bonds between different flakes and the presence of the carboxylated group on the surface allows the formulation of dispersions with a concentration of approximately 0.2 mg mL$^{-1}$ in methanol [28].

Based on previous works, a new surfactant was produced in situ by radical polymerization of diethyl maleate. This monomer was chosen because of its compatibility with ethanol and its tendency to completely decompose without carbonization at high temperature as is typical of acrylates. The reaction was performed in a microwave reactor to achieve precise control and uniformity of the temperature. The graphene concentration in ethanol reached 1.09 mg mL$^{-1}$ and the obtained suspensions stable over a long period of time, even months.

## 2. Materials and Methods

### 2.1. Materials

Natural graphite (flakes, 99% carbon basis, −325 mesh particle size, purity $\geq$99%), diethyl maleate (97%), 2,2′-Azobis(2-methylpropionitrile) (AIBN) ($\geq$98%), ethanol (99.8%), chloroform ($\geq$99%) and toluene (99.8%) were supplied by Merck. N-methyl-2-pyrrolidone (NMP) ($\geq$99.8%) was supplied by Carlo Erba Reagents. Nylon membranes (Whatman membrane filters nylon pore size 0.2 µm, diam. 25 mm) and alumina membranes (Whatman

Anodisc inorganic filter membrane supported, diam. 25 mm, pore size 0.02 µm) were supplied by Merck. The synthesis was performed using a microwave reactor Monowave 400 Anton Paar.

### 2.2. Synthesis

#### 2.2.1. Exfoliated Graphene

The exfoliated graphene (EG) was prepared by ultra-sonication; 1.000 g of graphite was added to 50 mL of n-methyl-2-pyrrolidone, the suspension was sonicated for 80 min at 10 °C with a frequency of 37 kHz and 320 W of power in an ice bath.

After the sonication, the solution was centrifuged at 4200 rpm for 45 min and the supernatant was pipetted away and stoked (Figure 1a). The presumable concentration was evaluated as 80 µg mL$^{-1}$. The solution was further sonicated before use as reported by Coleman et al. [29].

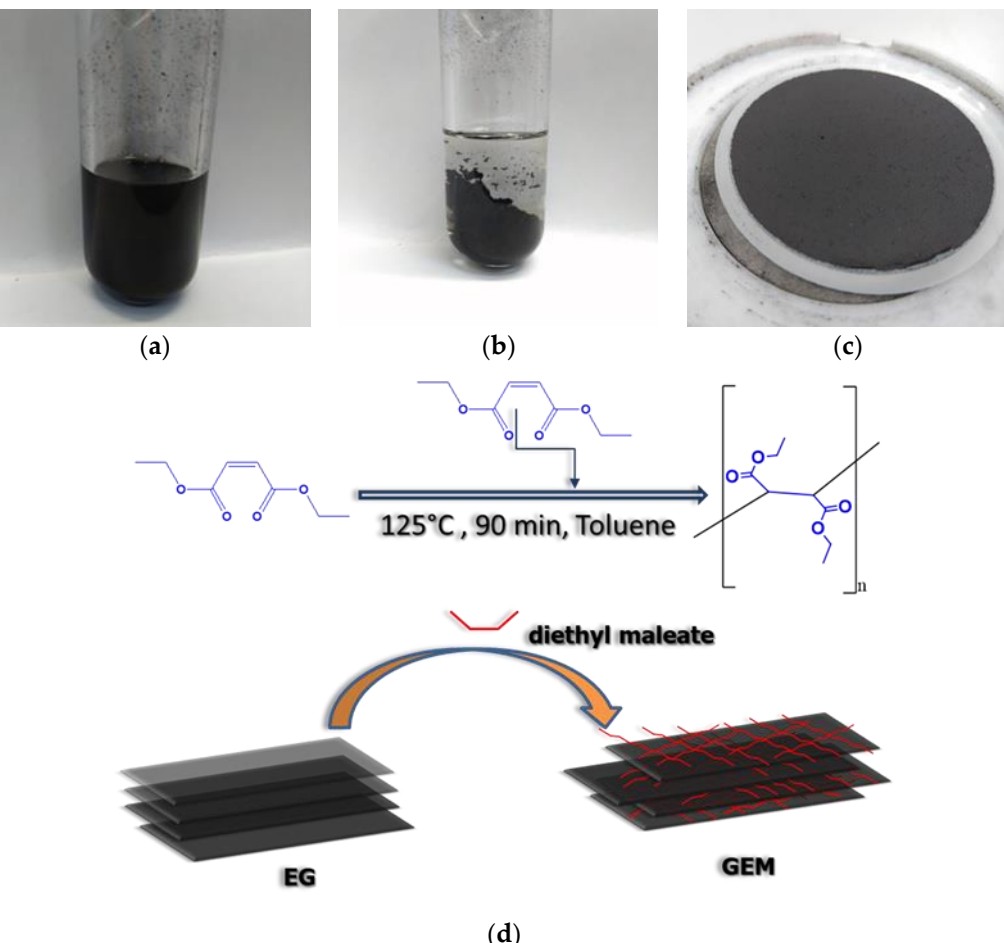

**Figure 1.** Graphene before (**a**) and after (**b**) microwave reaction. The sample was filtered with an alumina membrane (**c**). Reaction scheme and graphical representation of polydiethylmaleate on the exfoliated graphene surface (**d**).

#### 2.2.2. Graphene Diethyl Maleate

The EG was obtained by filtration on a nylon membrane using a vacuum filtration system. The graphene (about 30 mg) was washed with 50 mL of ethanol and thus re-dispersed in toluene. Then, 1.0 g (0.0058 mmol) of diethyl maleate was added to 10 mL of the toluene solution containing EG under a nitrogen atmosphere.

The obtained solution was reacted, using a procedure previously optimized, at 125 °C for 90 min in a microwave reactor (Figure 1b). The reactor was set to maintain a constant temperature.

The reacted solution was filtered on an alumina filter by using a vacuum filtration system. Thus, the solid filtrate was washed with ethanol and after with chloroform. The filter was dried in an oven at 60 °C under vacuum until it reached a constant weight. Graphene diethyl maleate (GEM) was then recovered (Figure 1c,d).

### 2.2.3. Graphene Diethyl Maleate Produced by Adding AIBN

The procedure for production of GEM was repeated, adding 0.190 g (0.00116 mol) of AIBN, a radical initiator, and a radical transfer. The graphene (about 30 mg) was washed with 50 mL of ethanol and thus re-dispersed in toluene. Then, 1.0 g (0.0058 mmol) of diethyl maleate was added to 10 mL of the toluene solution containing EG under a nitrogen atmosphere and 0.190 g (0.00116 mol) of AIBN equal to 20% mol of diethyl maleate.

The obtained solution was reacted, using a procedure previously optimized, at 125 °C for 90 min in a microwave reactor. The reactor was set to maintain a constant temperature.

The reacted solution was filtered on an alumina filter by using a vacuum filtration system. Thus, the solid filtrate was washed with ethanol and after with chloroform. The filter was dried in an oven at 60 °C under vacuum until it reached a constant weight. Graphene diethyl maleate (GEM) was then recovered.

### 2.2.4. Analyses

Thermogravimetric-coupled infrared absorption analyses (TGA-IR) were carried out in a thermogravimetric analyzer NETZSCH TG 209 F1 coupled by a transfer line heated at 230 °C with an infrared spectrometer Bruker TENSOR II equipped with an IR gas cell heated at 200 °C. The tests were performed by heating samples of about 3 mg in alumina pans from 30 °C to 800 °C with a rate of 20 °K min$^{-1}$, under nitrogen flux of 40 mL min$^{-1}$. Before the tests, three vacuum cycles were performed to remove solvent impurities on the surface of the materials and purge them of air. Experimental weight error is ±1%. The FTIR analysis was collected in the absorbance mode in the range 650–4400 cm$^{-1}$.

The morphological analysis was performed by means of a Zeiss Auriga dual-beam FIB-SEM microscope. Concerning sample preparation, EG and GEM samples were dispersed in ethanol and subsequently drop-casted on Cu lacey carbon TEM grids.

Raman characterization was performed by means of a Renishaw In Via micro-Raman spectrometer, equipped with a cooled CCD camera. A laser diode source (λ = 514.5 nm) was used with 5 mW power, and sample inspection occurred through a microscope objective (50×), with a backscattering light collection setup. Measurements were collected with a fast acquisition time (2 s) and were the average of 50 accumulations.

A PANalytical X'Pert MRD Pro powder diffractometer equipped using the 1D PIXcel detector was employed for the X-ray diffraction analysis (Malvern PANalytical, Malvern, UK). The patterns were collected in Bragg–Brentano reflection mode by using a Cu Kα1/2 radiation, at an operating voltage of 40 kV and a tube current of 40 mA. The instrumental broadening was determined by fitting PseudoVoigt functions to line profiles of a standard LaB6 powder NIST660c. The measurements were carried out in continuous mode with a step size of 2θ = 0.0131° and a data time per step of 150 s for the nylon membranes and 300 s for the alumina membranes. QualX software with COD-database was employed for the qualitative phase determination and MAUD free software for the quantitative analysis and the refinement. The COD cards matching the diffraction patterns are numbers 00-900-8569 (space group P 63 m c) and 00-901-2705 (space group R-3 m).

X-ray photoelectron spectroscopy was carried out with a PHI 5000 Versaprobe spectrometer (monochromatic Al K-alpha X-ray source, 1486.6 eV energy) over 500 μm × 500 μm areas of interest. The following pass energy values were used: 187.85 eV for survey spectra and 23.5 eV for high-resolution scans. A combined electron and argon ion gun neutralizer system was used for charge compensation during the acquisitions. The analysis of experimental data was performed with CasaXPS software. The binding energy scale was calibrated by placing the C-C sp2 component at 284.5 eV, using an LF (0.6, 1, 255, 350, 6) asymmetric line shape as provided in CasaXPS.

## 3. Results and Discussion

Figure 2 provides an investigation of the morphology of exfoliated graphene before and after modification with diethyl maleate by means of electron microscopy. By comparing low-magnification images of EG and GEM aggregates, the functionalization process does not induce significant morphological changes to the graphene flakes at the sub-micrometer scale.

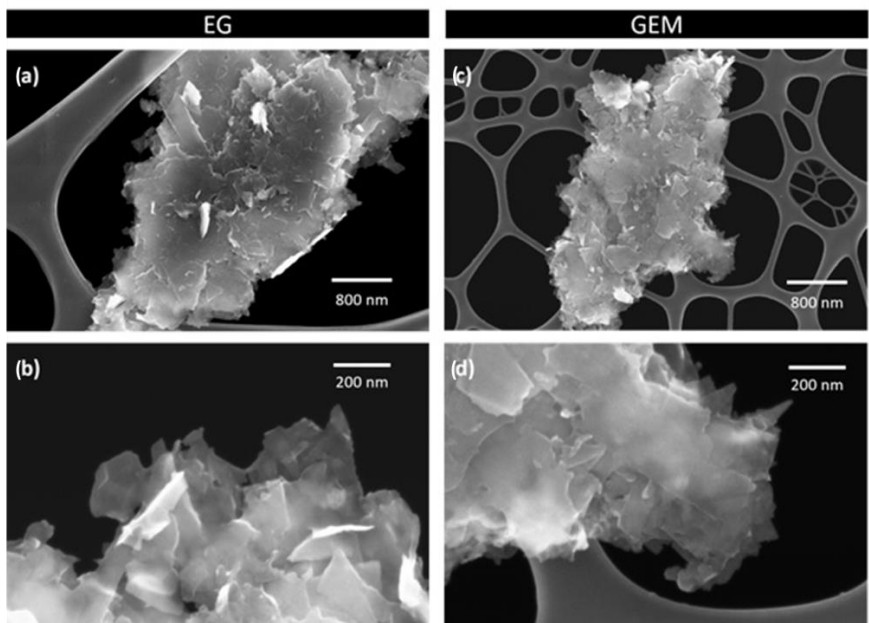

**Figure 2.** Field-emission scanning electron microscopy images at different magnifications showing exfoliated graphene (**a**,**b**) and GEM (**c**,**d**) on lacey carbon membranes.

A close observation of high-magnification images reveals that the morphology of graphene flakes is well preserved at the nanometer scale, with no evidence of induced nanoporosity in the basal plane. Moreover, it is interesting to notice that FESEM images suggest that the functionalization of the graphene flakes is spatially uniform, with no aggregation of diethyl maleate molecules at such magnification. To confirm the results, further images can be found in the Supplementary Materials Figure S4.

The TGA analysis of graphite (Figure 3) shows a continuous weight loss. Around 700 °C, the onset of a weight loss step is visible. The residue at 800 °C is around 96.7 wt.%. Water, $CO_2$ and a small amount of CO are produced (Supplementary Materials Figure S1) from oxygen impurities present on the graphite structure as clearly visible from IR analyses of the developed gasses.

Once exfoliated, the graphene shows a higher weight loss rate with a similar behavior, probably being the particles smaller they may be carried by the nitrogen flow, in addition to degradation. Similar to the previous sample, the IR analyses of the developed gasses confirms the production of water, $CO_2$ (FT-IR 750 °C), and a small amount of carbon monoxide that reaches maximum production at a lower temperature (SM Figure S2) due to oxygen impurities present in the sample. The different degradation rate of graphite and graphene are indeed correlated with the techniques used for the preparation of the composite; the sonication method [17,30] increases the oxygen surface functional groups.

In contrast, several degradation steps are present in the case of GEM (Figure 4a). The first step has the maximum degradation temperature at about 150 °C, the second at about 370 °C, the last step starts after 700 °C with the degradation rate increasing up to 800 °C. From FT-IR analyses (Figure 4b,c), the first degradation step shows the presence of several gasses such as water, CO, $CO_2$, and the presence of C-H stretching arising from alkyls. The degradation can be attributed to the de-alkylation of esters, which are present from the functionalization and the formation of degradation products and anhydrides.

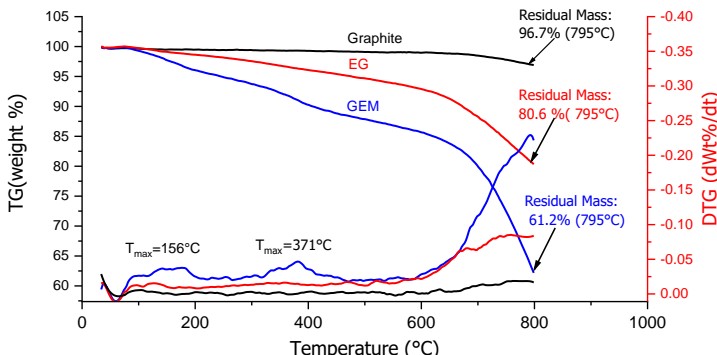

**Figure 3.** Thermal degradation analyses of graphite (black), graphene (EG) (red) and graphene ethyl maleate (GEM) (blue) weight loss (TG) and weight loss derivative (DTG).

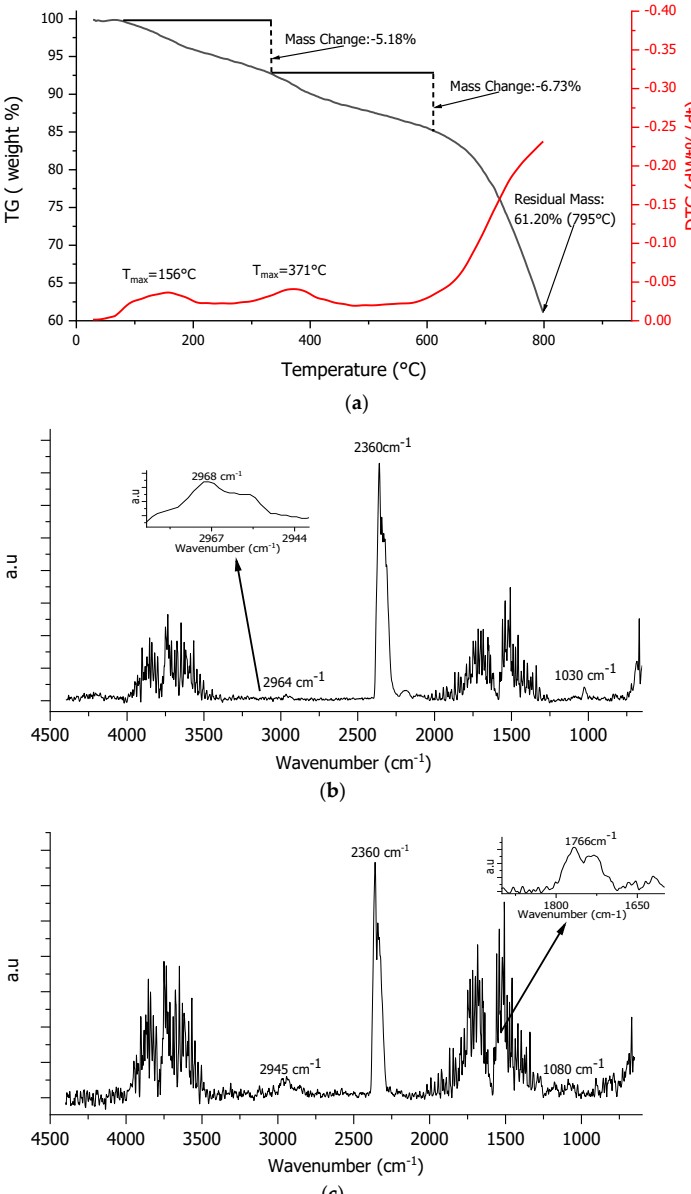

**Figure 4.** (**a**) Thermal degradation of graphene ethyl maleate (GEM). (**b**) Gas infrared spectra correlated with the degradation of GEM at 160 °C with (inside caption) the peak observed with the subtraction of water and (**c**) 380 °C with the peak observed with (inside caption) the subtraction of water.

The second degradation step at 370 °C shows the presence of the former gasses together with carbonyl groups and an acrylate double bond evidenced only after the subtraction of the spectrum related to water between 2000 and 1600 cm$^{-1}$.

It is possible to observe the double-stretching of C=O (1760–1725 cm$^{-1}$) that became the most prominent peak. The C–H rocking between 1030 and 1080 cm$^{-1}$ is clearly visible in the first degradation, but surprisingly the subtraction helped to discover the presence of this peak in the second degradation step as well and to confirm the continuous formation of esters derivate. Finally, it is possible to recognize the peak at 2945 cm$^{-1}$ related to the stretching of C–H of the ethyl groups of the diethyl maleate.

The change after the diethyl maleate functionalization is evident: the TGA confirms that the two degradation steps are directly related to presence of organic molecules.

In order to quantify the presence of the diethyl maleate, its amount was rationally increased by adding a radical initiator (azobisisobutyronitrile, AIBN) to GEM. This method has been reported in the Supplementary Materials. By adding AIBM, a mass loss starts at about 300 °C with a maximum degradation rate at 370 °C, just as has been observed in the pristine GEM sample and reported in Table 1. Indeed, AIBN increases the polymer formation. The GEM modified with the radical initiator showed a higher resistance to heating. As reported in Table 1, the ttal residual mass is higher than the weight of the material made without a catalyst. The radical initiator could also promote reticulation and, thus, preferential carbonization of the polymer.

**Table 1.** Influence of radical initiator and radical inhibitor in the reaction.

| GEM Preparation | Loss off Mass 370 °C (wt.%) | Residual Mass 795 °C (wt.%) |
|---|---|---|
| - | −6.73 | 61.20 |
| AIBN | −15.35 | 70.75 |

The comparison of the Raman spectra of graphene and GEM (Figure 5) shows that the exfoliation degree of the graphene is low, as demonstrated by the position of the G band near 1582.9 cm$^{-1}$, because the decrease in the number of layers of graphene increases the Raman shift of the G band [31,32]. The ratio between the intensity of the G band and the intensity of the 2D band is $I_{2D}/I_G = 1.44$, defining the starting graphene as a few layers [33,34]. The Raman spectrum of GEM is similar, demonstrating that the reaction with a microwave reactor has no influence on the exfoliation degree. The uncontrolled growth of the polymer also does not increase the number of defects on the surface of graphene, in fact the calculated ratio between the D and G band of GEM ($I_D/I_G = 0.48$) and the ratio $I_D/I_G$ of graphene ($I_D/I_G = 0.5$) is quite similar [34]. The lateral size ($L_a$) of the graphitic crystallites, therefore, according to Tuinstra and Koenig [35], is 9 and 8.7 nm for GEM and EC, respectively. Moreover, the absence of a broad peak at 1530 cm$^{-1}$ suggests that the samples have a high degree of crystallization and no amorphous carbon is present [36].

In accordance with the Raman spectra, the diffractograms of GEM and EG show similar features with a well-ordered structure and absence of amorphous broadenings. The spatial group has been identified as P63mc due to a very weak signal in correspondence with Bragg's reflection at 44.7° (2theta). The two samples show a strong preferential orientation around the (00l) planes and, as it is possible to observe from Figure 6b, the main reflections belong to the (002) and (004) crystalline directions. The values of the cell lengths, the crystal size, and density of the material according to the modulated turbostratic model are reported in Table 2, showing a good agreement with the estimation in the Raman spectra.

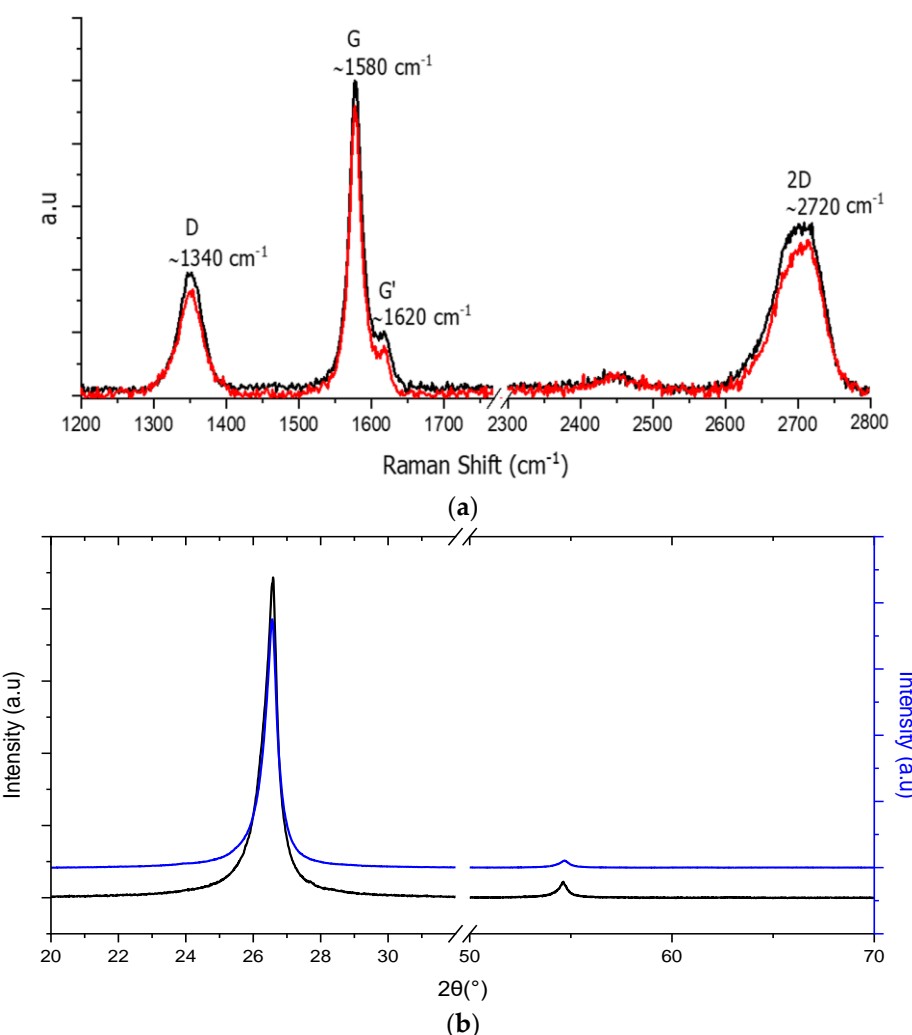

**Figure 5.** Raman spectra comparison of exfoliated graphene (black) and GEM (red) (**a**). XRD spectra of EG (black) and GEM (blue) (**b**).

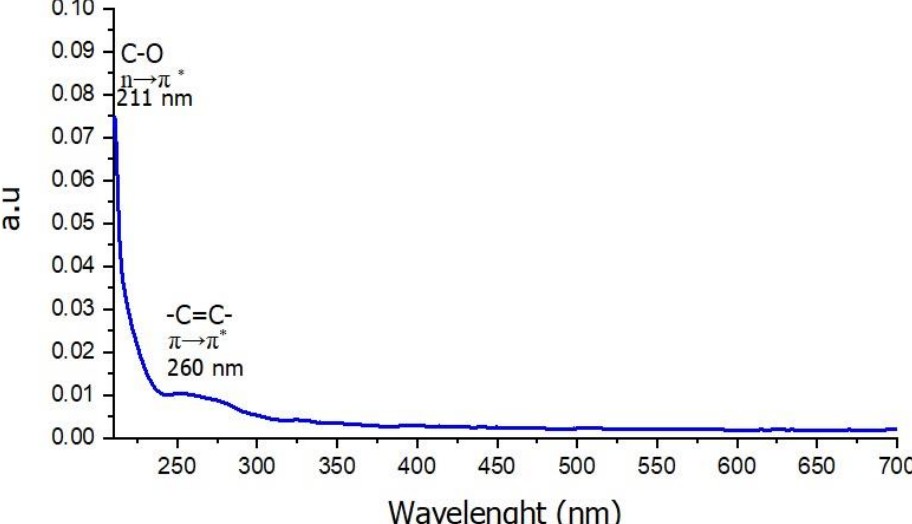

**Figure 6.** UV-Vis spectra of GEM.

**Table 2.** Parameter from XRD diffractogram.

| Parameters | | GEM | EG |
|---|---|---|---|
| Cell length [nm] | A | $2.17 \pm 0.05$ | $2.12 \pm 0.02$ |
| | C | $6.72 \pm 0.01$ | $6.71 \pm 0.01$ |
| Crystal size [nm] | | $7.53 \pm 4.76$ | $8.02 \pm 2.68$ |
| Density [kg/m$^2$] | | 2.90 | 3.04 |

The UV-Vis analysis showed three different absorption bands. The absorbance at 265 nm is correlated with the $\pi \rightarrow \pi^*$ of the -C=C- group of the aromatic rings of the graphene. The peak at 203 nm is correlated with the $n \rightarrow \pi^*$ of the C-O present in the esters group of the material, usually between 200 and 210 nm. It is worth noticing that with the increase in the concentration, the peak shifts to 200 nm.

The evaluation of the concentration was performed by the dispersing 30 mg of GEM in 10 mL of ethanol after 5 min of sonication. The dispersion was centrifuged for 5 min at 5000 rpm and the supernatant was isolated. The solution was diluted and analyzed with the UV-Vis spectrometer to estimate the maximum concentration of the material in ethanol using the Lambert Beer law:

$$A = \varepsilon l \, C \tag{1}$$

where A is the absorbance, l is the optical path, and C is the concentration.

The extinction coefficient of graphene in ethanol at 660 nm was used to calculate the concentration of the dispersion ($\varepsilon = 911.25 \, \text{L}^{-1}\text{g}^{-1}\text{m}^{-1}$). The calibration curve was reported in the Supplementary Materials (Figure S6), and we observed that the diethyl maleate has a negligible effect with the UV-Vis spectrum (Figure 6).

The maximum estimated concentration of GEM in ethanol is about 1.09 mg mL$^{-1}$.

Insight into the chemical composition of the exfoliated graphene before and after the modification with diethyl maleate is provided by X-ray photoelectron spectroscopy. Semi-quantitative analysis of survey spectra (see Figure S5 in the Supplementary Materials) immediately confirms the modification of the external surface of the flakes, with an increase in oxygen content (from 4.2 at% to 8.8 at%).

Careful analysis of the C1s region of the photoelectron spectrum provides further information on the functionalization process. Concerning pristine graphene, the C1s region (Figure 7a) shows the characteristic asymmetric C–C sp2 component, alongside the $\pi$–$\pi^*$ satellite, in accordance with the literature on graphene-based materials [37]. Low-intensity components related to oxygen-containing functionalities (C–O, C=O, COOH) are present, induced by the exfoliation process. After the modification with maleate, there is a slight increase in the C=O and COOH contributions, in accordance with the maleate molecular structure. Moreover, the $\pi$–$\pi^*$ satellite contribution (strictly related to the graphitic structure) is preserved, meaning that the functionalization process does not induce significant increase in defects, in accordance with Raman results previously discussed.

The solubility of graphene obtained using the method of Coleman et al. [29,38] is quantitatively compared by mixing the same quantity of graphene in different solvents as reported in Figure 8. The two solutions in DMF and NMP are clearly different from the other solution in solvents with a lower boiling point or with lower toxicity such as dimethylsulfoxide (DMSO).

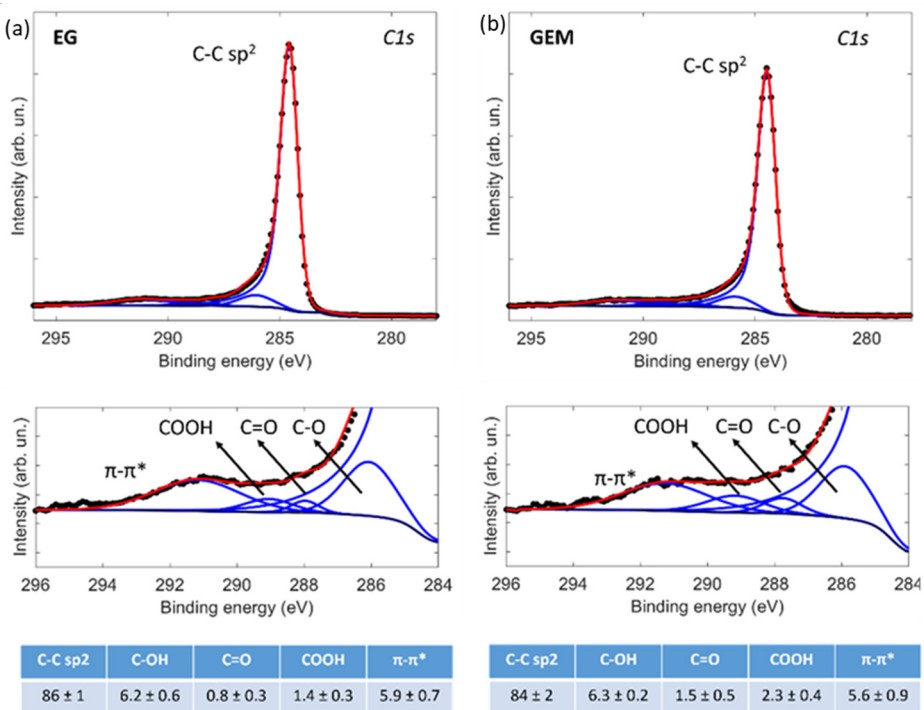

| C-C sp2 | C-OH | C=O | COOH | π-π* |
|---------|------|-----|------|------|
| 86 ± 1 | 6.2 ± 0.6 | 0.8 ± 0.3 | 1.4 ± 0.3 | 5.9 ± 0.7 |

| C-C sp2 | C-OH | C=O | COOH | π-π* |
|---------|------|-----|------|------|
| 84 ± 2 | 6.3 ± 0.2 | 1.5 ± 0.5 | 2.3 ± 0.4 | 5.6 ± 0.9 |

**Figure 7.** X-ray photoelectron spectroscopy high-resolution scans of the C1s region for exfoliated graphene (**a**) and GEM (**b**) samples, alongside results from fitting with synthetic components. Semi-quantitative results are provided, alongside uncertainties calculated with Monte Carlo routines in CasaXPS software.

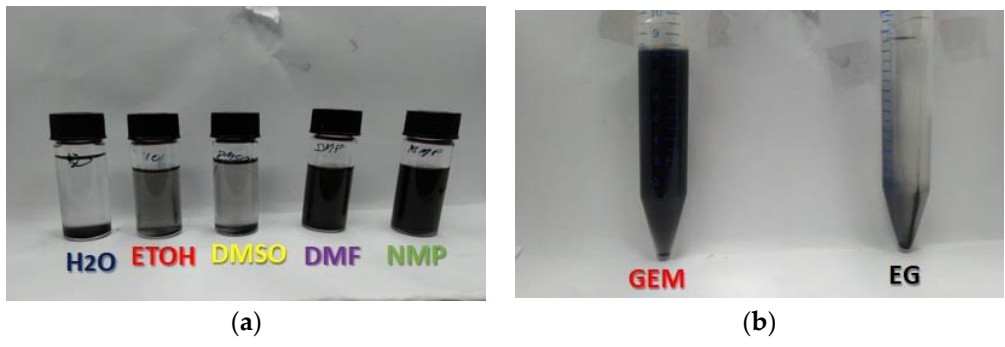

**Figure 8.** Graphene dispersion in a polar solvent (**a**). Comparison of a solution in ethanol of Graphene diethyl maleate (GEM) and Graphene (G). (**b**) Direct use of graphene in ethanol results in complete precipitation after 24 h (right) while a stable graphene dispersion in ethanol is obtained the solution is still stable after a year.

Instead, the result from GEM is readily soluble in ethanol; it dissolves by simply mixing and using a sonication bath for few minutes. Figure 7b clearly shows the difference in ethanol dispersions of GEM compared to graphene. Moreover, the solution is substantially stable for a very long time, reaching a year without meaningful precipitation after one year of storage in the dark at room temperature.

In Table 3, the concentration of ethanol solutions of different types of graphene materials are reported. Chen et al. prepared a graphene colloidal (GC) using p-phenylenediamine as a dispersing agent obtaining a concentration of 0.2 mg mL$^{-1}$ [39] and graphene composite with polyvinyl pyrrolidone (G-PVP) that gives the highest concentration of graphene dispersed in ethanol, reported in the literature [19–21]. The results provide evidence that the GEM material reaches a high concentration comparable with the benchmark in ethanol. However, GEM is probably more stable over time, with the concentration of

the other graphene materials starting to precipitate after few days/weeks in the other papers' descriptions.

**Table 3.** Comparison of concentration in ethanol of different graphene-based composites: GEM, GC (graphene colloidal), G-PVP (graphene with polyvinyl pyrrolidone).

| Material | Stabilizer | Conc. (mg mL$^{-1}$) | Ref. |
|---|---|---|---|
| GEM | Poly diethyl maleate | 1.09 | This article |
| EG | - | 0.09 | [29] |
| GC | p-phenylenediamine | 0.2 | [39] |
| G-PVP | Polyvinyl pyrrolidone | 0.8 | [19–21] |

The graphene suspension developed in this work can find different applications in the field of electronics. For example, this suspension can be used to produce stable graphene-based inks that tend to dry more quickly than normal solutions in high-boiling solvents for use in the field of photovoltaic cells [40], flexible electronics [41], and electromagnetic shielding [42,43].

## 4. Conclusions

In this study, a simple method to functionalize graphene and to obtain stable dispersions is presented. The use of diethyl maleate promotes the formation of a well distributed surface layer of organic functionalities that increases the affinity between graphene and ethanol. To the best of our knowledge, this is the first dispersion in ethanol that is stable for years and when the solvent evaporates, no deformation nor changes in the physico-chemical properties of the material were observed. Therefore, the GEM and the graphene show substantially identical properties, with the advantage of being able to disperse the GEM at a low boiling point and in sustainable green solvent, and to make inks stable for years. In addition, the GEM suspension has a maximum solubility of 1.09 mg mL$^{-1}$, which is further beneficial for reducing the usage of solvents.

**Supplementary Materials:** The following Supplementary Materials can be downloaded at: https://www.mdpi.com/article/10.3390/colloids6040075/s1. Figure S1: (a) Thermal degradation of graphite. (b) Gas infrared spectra correlated with the degradation of graphite at 380 °C. (c) The gas infrared spectra correlated with the final degradation at 798 °C. (d) The kinetics devolution of $CO_2$ and $H_2O$ during heating. Figure S2: (a) Thermal degradation of exfoliated graphene (EG). (b) Gas infrared spectra correlated with the degradation of EG at 380 °C. (c) The gas infrared spectra correlated with the final degradation at 798 °C. (d) The kinetics devolution of $CO_2$ and $H_2O$ during the heating of the sample. Figure S3: (a) Thermal degradation of graphene ethyl maleate (GEM) made with the adding of AIBN. (b) Gas infrared spectra correlated with the degradation of GEM at 220 °C with (inside caption) the peak observed with the subtraction of water. (c) 420 °C with the peak observed with (inside the caption) the subtraction of water. (d) The kinetics devolution of $CO_2$ and $H_2O$ during the heating of the sample. Figure S4: Field-emission Scanning Electron Microscopy images at different magnifications showing exfoliated graphene (EG) (a,b) and graphene modified with maleate (GEM) (c,d) on lacey carbon membranes. Figure S5: X-ray Photoelectron spectroscopy survey spectrum for exfoliated graphene and GEM, alongside semi-quantitative analysis, reported in the table. Uncertainties on relative atomic concentrations were calculated using Monte Carlo routines provided in CasaXPS software. Figure S6: Calibration curve of exfoliated graphene in ethanol. Table S1: Legend.

**Author Contributions:** Conceptualization, A.M. and S.B. (Sergio Bocchini); methodology, A.M. and S.B. (Sergio Bocchini); investigation, A.M., M.S., S.B. (Stefano Bianco), A.C. and M.F.; resources, C.F.P.; data curation, A.M.; writing—original draft preparation, A.M. and S.B. (Sergio Bocchini); writing—review and editing, A.M., M.F., M.S., S.B. (Stefano Bianco) and S.B. (Sergio Bocchini); supervision, S.B. (Sergio Bocchini) and C.F.P.; funding acquisition, C.F.P. All authors have read and agreed to the published version of the manuscript.

**Funding:** This research received no external funding.

**Institutional Review Board Statement:** Not applicable.

**Informed Consent Statement:** Not applicable.

**Data Availability Statement:** Not applicable.

**Conflicts of Interest:** The authors declare no conflict of interest. The funders had no role in the design of the study; in the collection, analyses, or interpretation of data; in the writing of the manuscript; or in the decision to publish the results.

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
