# Peer review of "Production of Graphene Stably Dispersible in Ethanol by Microwave Reaction"

_colloids, doi:10.3390/colloids6040075_

Round 1

Reviewer 1 Report

The authors report the functionalization of graphene by diethyl maleate polymerization and its dispersion into ethyl alcohol. It is an interesting manuscript; however, the writing of the manuscript, especially in methods and discussion-results, is inferior. For example:

Point 1. The title of the manuscript does not match the objective, methodology, and conclusions; I suggest changing the title. In addition, remove the new word from the title because ethyl maleate has been used in similar systems.

Point 2. I suggest making a figure of the functionalization/polymerization on graphene and its dispersion in ethyl alcohol.

Point 3. I suggest improving the writing of the manuscript objective (lines 82-88); in its current state, it is confusing.

Point 4. I suggest defining the acronyms GEM, EG, and all those used in the manuscript.

Point 5. I suggest improving the writing of the materials and methods section of the manuscript; in its current state, it is confusing. Adding subtitles would help to understand the manuscript better. In addition, write the section first the materials, then the methods and finally the characterization.

Point 6. The title of the article mentions graphene dispersible in ethanol; however, in the methodology used for washes, explain this, please.

Point 7. The reaction of diethyl maleate on graphene via microwave; how did you select microwave conditions? Why are time, temperature and conversion not reported?

Point 8. The polymerization of ethyl maleate via thermal initiation with AIBN, at what temperature was performed? How did you select the amount of ethyl maleate (0.0058 mol)? Is the final product dispersible in toluene? I'm considering this to be a second method, placing a subtitle; however, it is confusing because the discussion of results has to come from comparing both methods (microwave and polymerization).

Point 9. The results and discussion section has flacks when discussing the results, missing references. The total number of references, in particular this section, is 5 very low.

Point 10. It is necessary to add chemical structure characterization (FTIR, XPS) to visualize the changes in the chemical structure of graphene.

Point 11. In the polymerization method it is necessary to solubilize the functional polymer with graphene in solvent that dissolves the polymer that did not react with graphene. Subsequently analyze the solvent and washed graphene.

Point 12. UV-Vis calibration curve needs to be added.

Point 13. Improve figures and conclusions

Author Response

Dear reviewer, thank you for the constructive review. The work has been greatly improved. The English has been corrected, the layout has been corrected, and different analysis techniques have been added to prove the previously described results. In particular:

“Point 1. The title of the manuscript does not match the objective, methodology, and
conclusions; I suggest changing the title. In addition, remove the new word from the title
because ethyl maleate has been used in similar systems.”

The title was changed in “Production of graphene stably dispersible in ethanol by microwave reaction”

Point 2. I suggest making a figure of the functionalization/polymerization on graphene and its dispersion in ethyl alcohol.

Figure 1 was modified accordingly

Point 3. I suggest improving the writing of the manuscript objective (lines 82-88); in its
current state, it is confusing

The writing was changed from:

On the basis of these previous works, we studied the introduction of a surfactant by an uncontrolled growing in situ of polymers and oligo polymers based diethyl maleate. The monomer was chosen because of the ethoxy groups, the presence of a polymerizable functional groups and the tendency typical of acrylates to completely decompose without carbonization at high temperature. The reaction was performed in a microwave reactor to achieve precise control and uniformity of the temperature, since the power is tuned on the basis of temperature feedback.

To

Based on previous works, a new surfactant was produced in-situ by radical polymerization of diethyl maleate. The monomer was chosen because of the compatibility with ethanol and the tendency typical of acrylates to completely decompose without carbonization at high temperature. The reaction was performed in a microwave reactor to achieve precise control and uniformity of the temperature.

Point 4. I suggest defining the acronyms GEM, EG, and all those used in the manuscript

The following table was added in the supporting info:

Table 1S Legends

Name

Abbreviation

Exfoliated Graphene

EG

Graphene diethyl maleate

GEM

Azobisisobutyronitrile

AIBN

n-methyl pyrrolidone

NMP

Dimethyl formamide

DMF

Dimethyl sulfoxide

DMSO

Graphene colloidal

GC

Graphene composite Polyvinyl pyrrolidone

G-PVP

Point 5. I suggest improving the writing of the materials and methods section of the
manuscript; in its current state, it is confusing. Adding subtitles would help to understand
the manuscript better. In addition, write the section first the materials, then the methods
and finally the characterization.

The experimental part was modified accordingly

Point 6. The title of the article mentions graphene dispersible in ethanol; however, in the
methodology used for washes, explain this, please.

The ethanol was used to wash the graphene on the filter Anodisc 47 that has 0.1 micrometer size pore. Thus it is a standard procedure to filter on alumina substrate and recover by sonication (see experimental part Figure 1c)

Point 7. The reaction of diethyl maleate on graphene via microwave; how did you select microwave conditions? Why are time, temperature and conversion not reported?

The reaction conditions were reached after a long optimization based on qualitative observation of dispersibility that was not reported being developed in different time with different purpose and in a 2-3 year span of time.

The reaction is mainly a surface functionalization of a relatively small amount of graphene that it is estimated at the beginning thus the % of organic material on graphene surface is estimated by TGA decomposition while the conversion of diethylmaleate is of no use being the excess not deposited on the surface washed away by the previous step of washing on the filter (see Point 6). The duration and temperature of the reaction were added to the experimental part

Point 8. The polymerization of ethyl maleate via thermal initiation with AIBN, at what
temperature was performed? How did you select the amount of ethyl maleate (0.0058
mol)? Is the final product dispersible in toluene? I'm considering this to be a second method,
placing a subtitle; however, it is confusing because the discussion of results has to come
from comparing both methods (microwave and polymerization).

The AIBN was used to ensure the polymerization of diethyl maleate and compare the obtained material with the one obtained by simply polymerization using the TGA analyses, this test was performed after the whole methods was developed and thus the material was not studied deeply, and we do not want to develop nor propose a method based on AIBN. All the other parameters were the same of the previous test, the text of experimental part was changed accordingly in order to avoid confusion.

Point 9. The results and discussion section has flacks when discussing the results, missing
references. The total number of references, in particular this section, is 5 very low.

The discussion was modified adding further tests and describing the results on the basis of literature, several references were added.

Point 10. It is necessary to add chemical structure characterization (FTIR, XPS) to visualize
the changes in the chemical structure of graphene

XPs were performed and explained accordingly, XRD were performed added and explained. It is quite impossible to perform IR analyses of the sample because we only have diamond ATR infrared instrument and thus the absorption band of diamond cover very well all the intensities of carbon materials, however the XPS and XRD are significative.

Point 11. In the polymerization method it is necessary to solubilize the functional polymer
with graphene in solvent that dissolves the polymer that did not react with graphene.
Subsequently analyze the solvent and washed graphene.

It is practically impossible to separate the functional polymer from graphene, first of all because the quantities are very low as the experimental part suggest thus the possible polymer is lost in impurities, second because the interaction between graphene and these polymers and the graphene is really strong as demonstrated from the washing of the functional material with a solvent in which the functional polymer it is soluble. This strong bond (formed from several secondary bond and thus not covalent linked as confirmed from the analyses) it is the main reason because the graphene became dispersible in ethanol.

Point 12. UV-Vis calibration curve needs to be added.

TheUV-vis calibration curve was added in the supporting info file

Point 13. Improve figures and conclusions

Figures and conclusions were improved.

Reviewer 2 Report

Comments on colloids-1926092

The manuscript entitled “Synthesis of a new ethanol dispersable graphene based micro-composite material” presented the polymerization of (diethyl maleate derivate) on graphene obtained by sonication in a microwave reactor. The obtained material demonstrates good stability in ethanol also after a long time that can be used to deposit graphene by the production of inks or casting and easy removal of the solvent. At the same time, heating at 300-400 °C in an inert atmosphere allows the removal of the polymer to obtain pristine graphene with a low number of defects.

The manuscript is well-written, results are well-discussed, however, there are several amendments required to be resolved before accepting it for publication which are disclosed below:

·         The authors should highlight why they are focusing on Graphene in the Introduction. Since several other 2D materials show the same properties as Graphene such as MoS2.

·         The authors need to add some of their main results/findings in the last paragraph of the Introduction.

·         The authors are suggested to write their full name in their first presence. Please check that carefully, there are some abbreviations whose full names are not provided at the beginning. For example; EG in Line 108.

·         Please recheck the sentence in Line 129, “The filter was dried in a vacuum oven at 60 °C under vacuum…………..”.

·         The last paragraph of Section 2 is a bit weird in the reviewer’s pdf version. It is requested to recheck.

·         The organization of Figure 1 is not uniform, it is requested to organize in the same size and format. Similarly, the organization of Figure 4 is not uniform. Please recheck.

·         It is suggested to provide more FESEM images for better understanding.

·         The authors should cite some relevant references for the statement they made in Line 160.

·         The authors are highly recommended to provide high-resolution figures. For example Figure 3 is of poor resolution.

·         The authors should provide more discussions on the mechanisms for good stability in ethanol, and how can be used to deposit graphene by the production of inks or casting and easy removal of the solvent, which would be beneficial for readers to understand their significance.

·         The authors are suggested to add a few sentences about the prospective applications of their work in the last paragraph before the Conclusions.

·         It is also recommended to add some references from recent years of the related work. Some references are important to understand the progress of graphene-based composite materials and the advantages of 2D materials: Composites Science and Technology 189 (2020) 108022 and Materials Today Communications 31 (2022) 103858.

·         The language expression in the text needs to be carefully checked and revised. There are several grammatical mistakes.

Author Response

Dear reviewer, thank you for the constructive review. The work has been greatly improved. The English has been corrected, the layout has been corrected, and different analysis techniques have been added to prove the previously described results. In particular:

  • The authors should highlight why they are focusing on Graphene in the Introduction. Since several other 2D materials show the same properties as Graphene such as MoS2.

The first part of introduction was changed to underline why we are focusing on graphene.

  • The authors need to add some of their main results/findings in the last paragraph of the Introduction.

The main results were added in the last paragraph of the introduction

  • The authors are suggested to write their full name in their first presence. Please check that carefully, there are some abbreviations whose full names are not provided at the beginning. For example; EG in Line 108.

The abbreviations were checked, and full name were used at first appearance, The complete table of abbreviations were added in the supporting info.

  • Please recheck the sentence in Line 129, “The filter was dried in a vacuum oven at 60 °C under vacuum…………..”.

All the Materials and Method was changed.

  • The last paragraph of Section 2 is a bit weird in the reviewer’s pdf version. It is requested to recheck.

The last paragraph of section 2 was checked again.

  • The organization of Figure 1 is not uniform, it is requested to organize in the same size and format. Similarly, the organization of Figure 4 is not uniform. Please recheck.

Done

  • It is suggested to provide more FESEM images for better understanding.

More FESEM images were added in the supporting info

  • The authors should cite some relevant references for the statement they made in Line 160.

The graphite once exfoliated form very small particles as it is recognizable from the FE-SEM, in the presence of a flow it is highly probable that these particles may be carried away, in order to have a less confusing sentence the following phrase:

Once exfoliated, the graphene shows a higher degradation rate with a similar behavior, probably the exfoliation let the graphene defect more exposed to thermal degradation

Was changed with:

Once exfoliated the graphene shows a higher weight loss rate with a similar behavior, probably the smallest particles due to graphite exfoliation, in addition to degradation, may be carried by the nitrogen flow as small particles e.g. PM10.

Were the word degradation was changed with carried away to be clearer.

  • The authors are highly recommended to provide high-resolution figures. For example Figure 3 is of poor resolution.

Done

  • The authors should provide more discussions on the mechanisms for good stability in ethanol, and how can be used to deposit graphene by the production of inks or casting and easy removal of the solvent, which would be beneficial for readers to understand their significance.

The discussion part was modified, several techniques were added in order to confirm the results and the stabilization mechanism.

  • The authors are suggested to add a few sentences about the prospective applications of their work in the last paragraph before the Conclusions.

Few sentences were added about prospective applications before the conclusions.

  • It is also recommended to add some references from recent years of the related work. Some references are important to understand the progress of graphene-based composite materials and the advantages of 2D materials: Composites Science and Technology 189 (2020) 108022 and Materials Today Communications 31 (2022) 103858.

The reference from recent years were added in the few sentences about application with other references on other application before the conclusions.

Round 2

Reviewer 1 Report

The authors made the suggested corrections.

Reviewer 2 Report

The authors have answered all the comments. The manuscript can be accepted for publication.